# “It’s Important, but It’s Not Everything”: Practitioners’ Use, Analysis and Perceptions of Fitness Testing in Academy Rugby League

**DOI:** 10.3390/sports8090130

**Published:** 2020-09-18

**Authors:** Sam McCormack, Ben Jones, Sean Scantlebury, Dave Rotheram, Kevin Till

**Affiliations:** 1Carnegie Applied Rugby Research (CARR) Centre, Carnegie School of Sport, Leeds Beckett University, Leeds LS6 3QU, UK; B.Jones@leedsbeckett.ac.uk (B.J.); s.scantlebury@leedsbeckett.ac.uk (S.S.); k.till@leedsbeckett.ac.uk (K.T.); 2England Performance Unit, Rugby Football League, Leeds LS17 8NB, UK; Dave.Rotheram@rfl.uk.com; 3Leeds Rhinos Rugby League Club, Leeds LS5 3BW, UK; 4School of Science and Technology, University of New England, Armidale, NSW 2351, Australia; 5Division of Exercise Science and Sports Medicine, Department of Human Biology, Faculty of Health Sciences, The University of Cape Town and the Sports Science Institute of South Africa, Cape Town 7700, South Africa

**Keywords:** fitness testing, athletic development, rugby league, coaching, strength and conditioning

## Abstract

A plethora of research exists examining the physical qualities of rugby league players. However, no research has investigated practitioners’ insights into the use, analysis and perceptions of such fitness testing data that is vital for applying research into practice. Therefore, this study aimed to examine practitioners’ (coaches and strength & conditioning [S&C] coaches) perceptions and challenges of using fitness testing and the development of physical qualities. Twenty-four rugby league practitioners were purposefully sampled and completed a semi-structured interview. Interviews were transcribed and thematically analysed identifying five themes (it’s important, but it’s not everything; monitoring; evaluation and decision making; motivation; and other external challenges). The theme of “it’s important, but it’s not everything” emerged as a fundamental issue with regard fitness testing and the use of such data and that physical data alone does not inform coaches decisions. There appears conflicts between coaches and S&C coaches’ perceptions and use of fitness data, identifying complexities of supporting players in multidisciplinary teams. Collectively, the findings highlight the multifaceted nature of academy rugby league and suggest that practitioners should utilise fitness testing to inform player evaluations, positively influence training and assist with decision making. Moreover, practitioners should understand the combination of factors that influence fitness testing and work collaboratively to enhance talent development strategies.

## 1. Introduction

Fitness testing of athletes is commonplace within amateur and professional sports clubs [1,2], schools [3] and in the general population [4]. Strength and conditioning coaches (S&C coach) and sport scientists often administer “fitness testing batteries” for a number of different reasons. Fitness testing can provide information on an individual’s current physical performance (e.g., anthropometric, strength, speed and endurance qualities) [5]. Such fitness data can identify areas for improvement [6], and subsequently inform training practices [1]. Furthermore, regular assessment of physical qualities at specified timepoints can determine the effectiveness of training interventions and establish benchmarks and comparative data for positional groups and age grades [7,8]. In addition, practitioners utilise fitness data in decision making processes [9], planning training programme delivery [10,11] and talent identification strategies [12]. As such, fitness testing is an important element of the training process. 

Within rugby league, coaches and S&C coaches implement talent identification and development programmes with the overall aim of facilitating the progression of youth players through into senior professional status [13]. In doing so, players typically complete technical, tactical and physical training to elicit positive training adaptions [14,15,16]. Academy programmes are common practice across youth sports [17] in order to identify the next generation of players [18] and develop the physical qualities needed to progress through pathways [19]. Owing to the intense physical nature of rugby league match play, it is imperative for practitioners to ensure players are conditioned to meet the demands associated with match play [13,20]. As a result, objective markers of performance and physical development are routinely conducted [21]. 

Within rugby league, a plethora of research has investigated the physical qualities of youth rugby league players showing differences between playing levels [22,23,24,25,26,27]. Typically, superior anthropometric qualities categorise between playing positions [13], lower body power and muscular strength increase with playing standard [8,17,28], and greater aerobic endurance positively influences career attainment [17,29]. In addition, greater physical qualities are associated with enhanced sport-specific skills (e.g., ball carrying and tackling ability) [27,30]. Therefore, routine assessment of these qualities is of upmost importance to those involved in the physical development of rugby league players.

Although numerous studies have investigated the physical qualities of rugby league players, no studies in rugby league (or any sport to our understanding) have examined practitioners’ perspectives of the influence of fitness testing on player’s physical development and their coaching practices. A detailed insight into these perceptions could support and develop existing practices for rugby league practitioners to facilitate academy player’s progression, whilst allowing researchers to understand the field and stakeholders in this area. Fitness testing is important and regularly utilised by practitioners, although, how various practitioners use such data is unknown. As such, there may be potential challenges implementing and communicating fitness data effectively [10]. Limiting our understanding to objective measures of performance can potentially lose key information and perspectives from important practitioners (e.g., coach and S&C coach). Moreover, understanding rugby league practitioners’ perceptions on fitness testing could increase the impact and adoption of research findings in practice.

Understanding coaches and S&C coach’s perspectives on factors associated with physical qualities and fitness testing in academy rugby league is warranted. Such staff work closely with players daily, design working structures [31], determine training practices [10,32], and are involved in both selection and recruitment in professional academy rugby league clubs [13]. Practitioners typically engage in deliberate planning that can provide a sense of direction, monitor progress and identify individual responses of players [9]. In addition, these practitioners frequently conduct physical testing in order to monitor and develop such qualities [1]. However, coaches and S&C coaches may have varying perceptions of physical qualities, fitness testing and the use of fitness testing data, highlighting the complex, multifaceted nature of academy practices and the working of multidisciplinary staff teams. Therefore, the aim of this study was to explore practitioners (coaches’ and S&C coaches’) perceptions, use and analysis of fitness testing and the challenges associated with developing and evaluating physical qualities in academy rugby league players. Secondly, the study aimed to compare whether these perceptions differed between coach and S&C coach. This study would provide insight into rugby league coaches and S&C coaches’ perceptions of fitness testing that may enhance coaches’ practices within multidisciplinary teams and enhance player development.

## 2. Methods

To gain an insight into rugby league practitioners’ perceptions regarding specific topics, a mainly qualitative approach was used. The philosophical position of this study was informed by a pragmatic approach, which allowed the primary investigator to explore elite coaches perceptions from an applied and conceptual perspective [33,34]. Employing this philosophy allowed for the observation of current truths while understanding that opinions were likely to change over time, rather than definitive representations of realities [34]. Furthermore, this method attempted to establish solutions to current issues in the context of the study [35]. Therefore, the findings discussed in the current manuscript display the authors interpretations of rugby league practitioners’ perceptions. 

The research team have significant experience in working within an academy rugby league setting, which provided a considerable understanding of each context and associated jargon and informal terminology, and facilitated rapport with each participant [36]. Additionally, the research team’s in-depth knowledge of the club’s practitioners and players simplified the interview process. Moreover, being classified as an insider may promote genuine engagement from participants [31,37]. Transcripts and subsequent analysis documents were reviewed by the authors, acting as “critical friends” [38] and engaged in constant discussions which supported the development of eventual themes. Discussions generated feedback in how well each transcript was represented and ensured themes and interpretations were grounded in the data. Following critical discussions, all authors reached agreement on themes and data grouping [39]. Conducting a pilot interview with a rugby league practitioner enabled the “refinement and development of the research instrument” [40] affording the work to reach a level of rich rigour [36]. Consulting each participant through member checking during transcription and interpretation of themes similarly assisted in consistent trustworthiness of data. Finally, the primary investigator ensured critical thinking and self-reflection throughout to ensure sincerity of the process [36,41].

### 2.1. Sampling and Participants

Twenty-four male rugby league practitioners (mean ± SD, age: 35.0 ± 7.4 years; coaching experience: 10.0 ± 4.5 years) from nine rugby league academies in England participated in this study. Academy staff were chosen for this study given their role in facilitating player development prior to progressing to professional status. All participants were identified through their role as academy staff and in collaboration with the Rugby Football League. The participants were sampled purposefully in order to include different types of staff (i.e., coach, S&C coach) [42]. All participants were either the Head of Youth (e.g., academy manager; n = 3), rugby coach (n = 11) or S&C coach (n = 10). As such, given their specific roles in professional rugby league clubs, all participants were considered suitable and justifiably represented an expert sample. Of the practitioners who participated, 10 had an undergraduate degree in sport and exercise science or similar, and five held a master’s degree in strength and conditioning. One participant held a MPhil, whilst two more were completing PhD’s in strength and conditioning. Three coaches held post graduate diplomas in elite sport coaching. Thirteen coaches held United Kingdom Coaching Certificate (UKCC) level 3, while two coaches held level 4. Two coaches were certified with the United Kingdom Strength and Conditioning Association (UKSCA), and one was accredited with the National Strength and Conditioning Association (NSCA).

### 2.2. Procedure

Prior to all experimental procedures, ethics approval was granted from Leeds Beckett University research ethics committee (application reference 58776). The participants were contacted by the lead researcher and invited to participate via email. Practitioners from nine rugby league academies participated, representing 70% of academies in England. In order to achieve a deeper understanding of practitioner’s perceptions of the use and analysis of fitness testing, physical qualities and associated challenges with player development, semi-structured interviews were used. The interviews were conducted between April and August 2019 during the competitive season. 

Semi-structured interviews were conducted one on one with the practitioner in a private area and lasted 33.2 ± 6.4 min (range 31.3–46.7 min). A list of open-ended questions were used to guide the interview and explore participants perceptions of the importance of physical qualities and their use and analysis of fitness testing [39]. Prior to the commencement of the interview, the researcher provided a short briefing detailing the background of the study, information on the questions, the potential outcomes of the results, while also ensuring complete confidentiality and anonymity. The participants read the information sheet and provided their written consent prior to commencement of the interview. All interviews were conducted by the principal researcher. The semi-structured interview allowed the primary investigator to ask a number of pre-planned open-ended questions.

### 2.3. Semi-Structured Interviews

The interview guide was developed to obtain information on participants use, analysis and perceptions of fitness and physical qualities. The guide consisted of the following; monitoring of physical qualities (two questions; e.g., “How do you monitor the physical qualities of your players?”), the use of fitness data (7 questions; e.g., “Do you find fitness testing useful? If not, then why?”) and the challenges of developing and monitoring physical qualities (five questions; e.g., “What challenges do you face in developing, monitoring and evaluating the physical qualities of your players?”). Although the interview guide was not always followed in strict chronological order, it acted as a natural progression and allowed further questions to be asked when required. In addition, the same set of questions were asked to all participants, however, the researcher allowed the flow of the discussion to direct the questions which allowed participants perceptions to be explored [42]. Probing questions were used to provoke more discussion and to ensure participants had discussed everything before the researcher moved on to the subsequent question [43]. The questions were developed based on a review of the relevant literature as well as the research teams experience and understanding of the rugby league academy system in England. In addition, the guide was informed by guidelines for qualitative research and interviewing [42,44]. The interview guide was piloted with four expert coaches which resulted in several alterations to ensure that questions and terminology elicited information corresponding to the aims of the study. As such, the interview guide was deemed valid and reliable. All interviews were audiotaped and later transcribed verbatim according to each question in the interview.

### 2.4. Data Analysis

For closed response questions and descriptive data, frequency analysis was conducted to determine the percentage of participants who provided a response. Open ended question responses were analysed using thematic analysis [45]. Thematic analysis involves an investigation of participants experiences in relation to factors and processes that influence a phenomena [45,46]. This method was chosen as it allowed practitioner’s use, analysis and perceptions of fitness testing and physical qualities to be understood. Firstly, the interviews were transcribed and reviewed. The transcripts were read on numerous occasions to allow the researcher to become familiar with the data [47]. Following transcription and reading, codes were then generated inductively from the practitioner’s responses to highlight and label the primary aspects of the transcripts. Next, codes were placed into themes which identified the main concepts, and were reviewed regularly to ensure all data and relevant information was collected and epitomised adequately [48,49]. This resulted in the emergence of lower order themes which were categorised into sub-themes and higher order meta-themes [50]. The final two stages involved naming and defining the themes and providing descriptors of each theme. Links were also made to the research question and relevant literature. This was primarily carried out by the lead researcher who engaged with the research team in constant discussion.

### 2.5. Findings and Discussion

#### 2.5.1. Fitness Testing Use

Table 1 presents practitioners responses to closed questions regarding fitness testing data, facilities and staff.

#### 2.5.2. Practitioner Interviews

Five themes were identified that represented rugby league practitioners’ perceptions of use, analysis and perceptions of fitness testing: “It’s important, but it’s not everything” was identified as the main higher order theme, with “monitoring”, “evaluation and decision making”, “motivation”, and “external challenges” making up the remaining themes. A thematic map is shown in Figure 1 to visualise the themes. Novel insights are presented, providing information pertaining to the use, analysis and perceptions of academy rugby league practitioners.

#### 2.5.3. It’s Important, but It’s Not Everything

This high order theme is based around practitioner’s perception of fitness testing which is; that fitness testing is important, but numerous other factors must be considered within academy rugby league. Coaches perceived that fitness testing was useful at certain timepoints for various reasons including “decision making”, to “benchmark players” and provide “a comparison”, however, it also depended on other factors (e.g., technical/tactical skill). The findings support several investigations into the importance of physical qualities and the use of fitness testing in academy rugby league to differentiate between playing levels and positional groups [16,51], inform training practices [5] and how superior physical qualities positively influence career attainment [1]. Overall, rugby league coaches interpreted fitness testing as useful, however relied on various other aspects of performance (e.g., rugby skills, psychological skills) for player development, which is not surprising given their primary role is the technical/tactical preparation of players [10]. Previous research has shown superior sport-specific skill as beneficial for selection in academy rugby league [52], and how physical qualities influence on-field performance [27], which support coaches viewpoints. Player selection is largely the coach’s role and fitness testing and physical monitoring seem to have a small part to play in these circumstances.

One coach spoke about the benefit of fitness testing to inform training programmes, however highlighted not to rely solely on results when making important decisions regarding team/player selection and progression. He also discussed the individual variability that is often apparent within fitness testing and whether this fully represents rugby performance:

“It’s (fitness testing) good at certain stages of the season; pre-season especially for working out loads etcetera but other tests can be misleading on how a player actually is. Some players will play a hell of a lot better than what their testing will show. So, there’s always going to be players that are just good at rugby and just switch off at (gym) training. Now they won’t switch off so that you bollock them but they won’t concentrate or exert themselves. But then you put them on a field, and you’ll be like “how’s he done that”. So, as a coach, it’s very important, just like it is with any other discipline not to just put all your eggs in one basket with fitness testing. Especially at our level, there’s too many players that get progressed through because they’re showing really good at testing, likewise if there’s a few other kids who get f****d off because they’re not strong enough”

Coaches often cited how other factors (e.g., rugby performance, injuries, the individual) must be considered “in conjunction” with fitness testing data, but also indicated the importance of certain physical qualities for rugby league performance. The coach also highlights S&C coaches’ potential difference of opinion:

“It’s (fitness testing) not the be all and end all, it’s in conjunction with… on the pitch. A number of other factors… you’ve got to take the individual, I suppose injuries (and) things like that, if you’re injured, you’re not going to perform to your max, there’s a lot of other factors. I suppose the big one from me, the S&C might have a different view; is that it’s in conjunction with other skills and performance and that. I suppose there are those unique players that the stats don’t particularly reflect their performance. It’s an important component in any sporting environment, I suppose if you’re not fit enough or strong enough then you can have the most skilful game but then after 20 min of the game you’re walking around and you can’t repeat the efforts then it doesn’t really matter”

The following coach stated his outlook on the use of fitness testing data which highlights the notion that fitness test performance is “not a good indication of match performance”. In addition, he discussed the individual nature of fitness testing and rugby performance, highlighting that fitness testing does not adequately recognise all aspects of performance:

“I use it (fitness testing data) to make decisions, absolutely, team selection too. But as you know yourself, there’s guys who are strong in the gym and not strong on the pitch and vice versa. There’s always been people like that. But for me, it’s as long as you are strong on the pitch”. 

This theme is further supported by another coach stating that he uses fitness testing to “help compare athletes” and obtain information on player’s fitness levels, however indicated that it would not make up his final (selection) decision; “It wouldn’t be the final say, it would contribute towards selection but would not make my mind up for definite”. Fitness testing data is used for a variety of reasons; to design player’s training programmes to suit their individual needs and ensure they “keep on top of things”, “to make sure that they’re at where we need them to be” and “set targets for kids to come back and things like that”.

There were some differences observed between coaches and S&C coaches regarding fitness testing. As expected, S&C coaches perceived fitness testing as useful and regular utilise various methods to monitor and evaluate the physical performance of players [10,32,53]. One S&C coach mentions how he uses fitness test results for comparison purposes, to generate training programmes, whilst also ensuring the training is beneficial for players: 

“(fitness testing) allows us to compare athletes of certain age groups and positions, and it helps guide us and inform our training regime. It also gives us something to take to coaches and allows us to check the effectiveness of our training programmes.”

Another S&C coach identified the benefits of fitness testing to “plan for particular player’s needs” and “group players for training programmes” while the use of subsequent data “has informed what we (S&C) have done, definitely”. 

In summary, practitioners found fitness testing to be useful for various reasons (e.g., training prescription, player comparison), indicating that both understand and value its use in practice. However, its importance is often variable between practitioners. Coaches perceived fitness testing to be useful (67%), however felt that performance on the pitch was more important regarding playing performance and selection (team selection, career progression). The majority of coaches utilised fitness data when making various decisions regarding player development and “making things quantifiable”, though, indicating the highly individual nature of various tests and players. Coaches identified that some players strengths may not be evident throughout fitness testing but can perform adequately on the pitch. Moreover, some coaches highlighted that some players “can blitz fitness testing and then can’t turn up to play in a game”, which further exacerbates the main theme. Numerous coaches stated how some player’s rugby performance may not be reflected wholly by their *physical* performance via testing results, and vice versa.

In contrast, all S&C coaches perceived fitness testing as positive and influential for their role. They typically used fitness testing to “regularly screen the boys” to “allow before and after player comparisons to demonstrate progress against their former self”, and to “give a snapshot of qualities that need to be improved and prioritised”. S&C coaches viewed the use of fitness testing and data to improve player preparedness, while coaches tended to focus on their responsibility; *rugby* performance. Moreover, the between-practitioner differences highlight some of the difficulty’s rugby league coaches and S&C coaches face in player development. 

### 2.6. Monitoring

This theme surrounds practitioners’ perspectives on the monitoring of physical qualities during the season. The theme of monitoring was associated with further sub-themes of “questionnaires” (e.g., wellness questionnaires), “observations” (e.g., watching training), “coach discussions” (e.g., speaking to the S&C coach/assistant coach/player), and ‘fitness testing’ (e.g., planned physical testing at specific timepoints). Practitioners typically employed an array of methods (e.g., wellness questionnaires, fitness testing, fatigue monitoring) to monitor physical qualities and performance of players consistent with previous literature [10,53]. Observation was recognised as coaches preferred method of analysing player’s performance, which is in accordance with research carried out in rugby union and youth football where “direct observation” and “coaches’ eye” were the most popular methods of monitoring players [53,54]. 

Coaches primarily monitored the physical qualities and performance of their players through observation and coach discussion. The following coach cites how he routinely conducts training sessions where he can observe players performance. In doing so, he can gather certain information regarding a player’s technique. The coach also states the individual nature of performance, and how player’s performance may not be reflected through objective measures such as load lifted in the gym:

“Observing wrestle and contact (sessions) which I do regular(ly). I’ll take these sessions; I can see if someone’s been put on their back. When I watch, I can see if its (the) technical aspect of the wrestle, your levers, your hips, your weight, where your head is or it’s a physical one. We had one kid last year who wasn’t strong enough and he was getting b****ed. So…. then I had another kid who wasn’t that strong in the gym, but he was that technical (he could do well). When I watch someone wrestle, you’d presume they were strong in the gym, but then you watch them, and they can’t lift a weight (in the gym)”. 

One coach explained a sport-specific drill that is employed to evaluate and “test” player’s performance, and how he can make a judgement on their ability to play high-level rugby league:

We actually set up drills specifically to see whether someone has got not just physical strength but the actual attitude. We get four (tackle) shields so it’s really close, and they ramp it up through them. That’ll tell me if someone’s got like that mental and physical strength that’s needed to be a rugby player. I can tell within five minutes if somebody’s going to be any good, I’m not being big headed but if you run into those shields and you’re cowering off a little bit then you’re not going to make it (in professional rugby league). The game’s that easy, you run hard, you tackle hard”.

One coach perceived fitness testing as useful, however, was unsure of what some of the test results meant; “I’d like to know what to look for and why… I would expect the S&C to be able to translate the numbers and tests into layman’s terms for me… I want to know the purpose of what they’re doing and why”. This highlights the need for simple interpretive data from performance measures to ease decision making and talent identification processes for coaches.

The sub-theme of “coach discussion” was evident when conversing about coaches’ methods of monitoring the physical qualities and performance of players. Typically, coaches monitored players qualities through speaking to players individually or with other members of staff (S&C coaches, physios, other coaches etc.) “It’s through talking to players, they’re normal people, it’s what normal people do”. Another coach cited how he’s in regular contact with the S&C coach who informs him of players performance:

“I’ve nothing to do with them in the gym. I don’t really monitor so I don’t use this. What I say to [S&C coach] is “how is it going with the strength at the beginning of the year? Are there any concerns?” We’ve had players who are losing weight so we’re saying; “what’s going on here”? So, I usually just speak to [S&C coach] if (there are) any concerns. We have a meeting on a Monday, and we all have our little sections for what we speak about. So [S&C coach] will say “I’ve got a concern with this or what do you think about this”. He’ll [S&C coach] tell me what volume he wants of a typical week. I don’t say “I want all this time” I say to him “right, you tell me what you need, and I’ll work around you” it’s worked really well so far”.

Typically, S&C coaches use fitness tests and “readiness to train” markers at various timepoints to monitor and evaluate physical qualities, similar to research in rugby union [53] and professional soccer [32] which showed that training load and fitness and fatigue is routinely measured by S&C coaches. Further sub-themes of “questionnaires” and “observation” were identified as methods used by S&C coaches to monitor the physical qualities and physical performance of players. One S&C coach stated the importance of observing players on the pitch as well as in the gym, which is important in order to understand what qualities some players need to develop e.g., endurance, acceleration, power. The quote again typifies the notion of individual variation in fitness testing performance:

“Observing in the gym, their loads, technique. I think observing in matches (is important), watching the games, sometimes you might see a lad doing really well in testing, but actually in a game situation suddenly (he) doesn’t seem to have the legs. So, I think that is then good feedback, so it’s always observing”.

One S&C coach explained how he monitors physical qualities and performance using “regular daily monitoring of wellness and physical markers… then just your typical general monitoring by looking through players programmes and having chats with them”. Another S&C coach highlighted the importance of league-wide fitness testing to provide information on player’s physical performance in comparison to national averages.

It’s clear that there are some differences and similarities between coaches and S&C coaches’ perceptions of monitoring physical qualities, fitness testing and performance, which is similar to findings from professional football [10]. Weston (2018) found that coaches and fitness staff (S&C coach, sport scientist) had a lack of agreement on the use of training to enhance fitness, however, agreed on other variables such as training load monitoring. Both coaches and S&C coaches in the current study cited observation as a method. However, coach’s observation was typically “using their eyes” and observing players rugby performance on the pitch, with one coach stating “I use my eyes a lot. It’s very subjective. It’s with your eyes really, isn’t it?”. This data is similar to findings in football where coaches’ subjective ratings of performance were valid and reliable at rating players’ potential [54,55]. Moreover, due to the multifactorial nature of rugby league, coaches routinely identified that objective performance measurements often do not effectively assess athlete’s sporting ability. In contrast, S&C coaches typically employed objective measures of observation, which included wellness questionnaires, daily and weekly monitoring, and global positioning system (GPS) data. This is to be expected given coaches and S&C coaches’ different roles within the team. 

GPS emerged as a constant topic of conversation from both coaches and S&C coaches and is a sub-theme of monitoring. The majority of both coaches and S&C coaches indicated the use of GPS technology within their practice. In general, practitioners utilised GPS to obtain information on training loads and match play characteristics; “It helps us analyse and improve those individuals”. While also informing training, generating training reports, and influencing decision making. One coach cited the importance he places on GPS data; “we’re massive about GPS data, I find it really helpful. Every training session we do, (sport scientist) sends me an update about what we do, its brilliant”. And also stated how such data can highlight certain players who may not be performing sufficiently; “I can see the forwards in one chunk, the hookers in another and so on… I can see an individual who runs one kilometere less than everyone else, why?” Our findings are in agreement with research in professional soccer, whereby practitioners use GPS equipment at every training session [32]. The use of such methods in rugby league has helped establish match demands [56,57] and informed training practices [14], advocating its use for monitoring purposes in academy rugby league. 

Furthermore, all practitioners stated they would welcome greater GPS use. Typically, clubs have a limited number of GPS units and access to these are limited; “ideally, we’d like to GPS the full team to provide feedback on all positions”, with funding being the main reason behind this. One coach stated the difficulty of not having a full set of GPS units to monitor training and provide information; “I’d like to use more GPS data; we’re guessing at the minute. I’ve come from a background where I’ve been told how many metres we’ve ran, how much high-speed running…” 

Interestingly, coaches perceived GPS data to be more useful for monitoring and evaluating player’s physical performance than S&C coaches (47% vs. 33%). Some S&C coaches stated that they use GPS technology, however, the sub-theme of GPS typically emerged from discussion with coaches. The main higher order theme of “it’s important, but it’s not everything” was also evident with one coach identifying some of the challenges that arise between coach and S&C coach regarding GPS technology to monitor and evaluate training and performance:

“My opinion as a coach vs S&C coach. If I’m told by sport science team that were doing three kilometres today, and the sessions been poor and we’re approaching that threshold where we’ve been told to stop, then I’ve got to get that balance right knowing that I don’t want to disregard what (sport scientist) is saying. But, (if) I’m not happy with that session, (then) we need to do a bit more. I use sport science as a bit of marker where to be but it’s not everything. I’d use it as a bit of a guide and I stick to it as much as I can, but the odd time (the sport scientist) might give (me) thumbs up through (the) window, and I’ll say “we’ve got a bit more to do just yet” because we need to do this so… so that’s the challenging part”

Interestingly, coaches reported using GPS more than S&C coaches, which may be attributed to coaches implementing GPS metrics to provide information on match demands and as indicators of performance. Moreover, coaches are responsible for the planning and designing of training and may require such information to determine training load. Although GPS was a frequent topic of conversation, it is not classed as fitness testing.

### 2.7. Evaluation and Decision Making

The theme of evaluation and decision making can be further divided into sub-themes of benchmarking players (e.g., setting standards), comparison (e.g., comparing to the national average/positions/levels), informing training (e.g., extra conditioning), and selection (e.g., team selection). Decision making has an integral role to play in coach’s day to day practice [9] and is associated with coach expertise [58] and successfully achieving their goals [59]. Moreover, decision making is equally as important for S&C coaches in order to make decisions and implement their practices efficiently [9]. Overall, practitioners utilise fitness testing data to benchmark players against other positions, standards and their competitors. It is also typically used to inform their decision making and training practices. “I just need the information to know if a player is fit to train or play, and for training numbers so I can plan my sessions” and “I might use it (testing results) to make decisions based on what players need” were quotes that symbolised a number of coaches perceptions regarding using fitness data. One coach cites how he uses fitness data to make specific decisions:

“If I had a kid at U19 that was really strong but not the quickest and I wanted to get him powerful I’d obviously work on speed development. Again, if we got a kid that’s showing up really fit but he’s lacking on other things… there’s going to come a point in time where we know that if we don’t sort him… get him stronger, we’re going to bottleneck him. Let’s say if I’ve got an 18 year old and we like everything about him, but I’ve got a first team coach who’s going to me “he’s not big enough for what I want” obviously we need to then worry about… “right we need to do a bit of hypertrophy” so they fill out a little bit. So that’s when that will come in”

Another coach stated how they use fitness data to inform training based on players strengths and weaknesses; “The blocks of training are focussed on players individual needs. Training can be tailored to where they need to improve”. All S&C coaches used fitness data to “prescribe training”, “to inform where an academy player is in relation to the 1st team and then identify weaknesses based on this”, and also “to set a benchmark that players must attain to progress to the next level”. Although some coaches previously stated during the interviews how they did not utilise fitness data or find it useful, all coaches cited that in fact, they regularly do use fitness data to make important decisions and prescribe training, which is in contrast to their previous opinions. All S&C coaches generally used such data within their practice. This philosophy aligns to Till’s decision making framework that using fitness testing data increases our understanding of players that can inform our planning, delivery and reflections of coaching [9].

### 2.8. Motivation

The theme of motivation constantly emerged throughout all interviews with practitioners. Motivation can be further divided into sub-themes of “competition” and “pushing each other”. Practitioners perceived fitness testing to be a useful tool “to set goals” and to identify “areas for improvement” for players in order to improve both performance and fitness testing results. Moreover, practitioners can manipulate training practices to ensure a competitive environment. In doing so, coaches can encourage development by facilitating competition amongst players. Research in soccer has advocated the use of “challenging” environments to develop psychological skills such as mental toughness [43,60]. One coach cited how they typically use such fitness data:

“(we use testing data in) goal setting, development, encouragement and as a motivation tool I suppose for all the boys to sort of reflect as well and look at their data. See how they’re doing themselves and areas they need to improve on, areas they thought they were better at. If a player is falling off standards, then you can go back to analysis from 3 months ago and show… You can set goals for 3-month periods and if you’ve not achieved them you can see”

One coach stated how he would allow players to see the national fitness testing results in order to motivate and improve players, which is also linked to comparison:

“It’s good for players to see what they’re up against. They get to see the team’s results. At the end of the day, you’re in competition. That’s professional sport, there’s literally no boundaries to where the game goes. I’d open it up to show what others (clubs) have got. I’d let every single player show their data, some players won’t give a f*** they’ll just be worried about themselves. But then all of a sudden there’s someone here who thinks they’re the bee’s knees, who’s s*** hot then all of a sudden there’s someone bigger than me, stronger and faster than me, then I think that would give them a bit of a drive”

Another coach described how important the use of fitness testing data was for players daily motivation:

“I’m a massive fan of the traffic light system, so green is good. So, in the gym for example, we’ve got a sheet, it’s there every day, they’re seeing it all the time, staring at it. So, when they’re showing up and they’re thinking “I’m not really feeling it today”, well, you’re at f*****g amber son, you better be feeling it. Simple as that”

Given that players are in a “competitive environment”, typically, all practitioners use fitness data as a method of motivation and to “give players the edge” when, they’re all “fighting” for positions and “pushing for that place”:

“Everyone gets to see (fitness test results), (they’re) up on a board. There’s a bit of that from a psychological perspective; If you’re bang average then you need to know you’re bang average and see what good looks like”

Practitioners typically stated how they make test results public and group players by positions/scores to enhance competition between players. Research has shown that factors associated with motivation such as being dropped from the team, facilitated soccer players transition to professional levels [61], which promotes practitioners’ use of these methods. A number of coaches in the present study cited how they post results in “WhatsApp groups” and “around gyms” for players to “see what level they’re at, it shouldn’t be a secret” in order to foster competition:

“We’re in a competitive environment, let’s be honest, ‘I wanna get the spot over you’ so we put them in positional groups so they’re against each other”

In general, both coaches and S&C coaches utilised fitness testing as a motivational tool “to be pushing each other” and to “add a little bit of peer pressure”. Highlighting its importance, the practitioners conveyed how fitness testing and results can be used to encourage motivation and competition amongst players to ultimately, improve overall performance. 

### 2.9. External Challenges

The final theme of external challenges typically arose from a number of factors that were out of practitioners control and included sub-themes of funding, time, parent education and control. The lower order theme of funding was cited regularly by practitioners and was one of the main challenges faced when developing, monitoring and evaluating the physical qualities of academy rugby league players. Coaching staff at some clubs were “part-time employees”, with other clubs “sharing training venues and equipment with other teams”. Inadequate club infrastructure and resources were identified as constraints soccer practitioners faced when monitoring training load [10], which are similar to our study as funding for GPS technology was regularly cited as a difficulty that practitioners regularly face.

Time was identified as an influential variable in academy rugby league player’s development, and a number of factors associated with time were seen to be disadvantageous for practitioners. Typically, S&C coaches spoke about their lack of time with players to develop particular physical qualities, and also time to discuss topics with coaches and to analyse fitness data. As one coach explained “the contact time we have with players is very limited, everybody wants a piece of them” and “our training time is sometimes cut short as we share facilities with the first team, so we are not a priority”. 

Moreover, time, or patience was mentioned on a number of occasions by coaches with regard players. Coaches felt there should be a sense of “patience” within a rugby league academy. Further, coaches were of the opinion that they need to be more conscious of the different player development rates, and not disregard players based on their physical qualities at a young age, with one coach stating:

“Everybody develops at a different rate so we’re talking about… we’ve got to learn not to write someone off too early. So, in the first year as a 19, just because you’re not big enough, fast enough or strong enough doesn’t mean you’re not going to be. It’s about being patient and taking your time with the player. So, I think patience and timing is the key with that”

All S&C coaches identified time as a challenge faced in developing physical qualities of players. S&C coaches typically referred to time as “recovery time”, but also “time with players” and indicated that sometimes there was a lack of sufficient recovery time following training, impacting on “players readiness to train” and as a result “decreases the value of what you are getting”. The notion of “finding time to plan with coaches” and “finding the time to carry out detailed analysis of data” was also identified by a number of S&C coaches as challenges in developing and evaluating physical qualities. 

The next factor of parent education was typically discussed by coaches as difficulties they faced during player’s development. Parents interfering with programmes was mentioned by coaches and was viewed negatively when parents were perceived to question a coach or their programme; “his dads just taking him to the gym on the evening, like so basically f*****g up the programmes we’re doing with him on the morning”. It appears coaches are wary of a number of external challenges, however, placing heavy restrictions on players may negatively affect both their performance and personal development. One coach stating “parents need to be more educated and stop kids from overtraining, I think there’s too many that don’t understand proper athletic development” with another coach citing how a parent interferes with their training programme. 

The final sub-theme of control is linked to external challenges for practitioners and is associated with “external commitments” (e.g., to other teams), “player adherence” (e.g., adherence to advice) and “athlete buy-in” (e.g., players wanting to train). Generally, coaches and S&C coaches felt they were unable to control player’s external commitments as “it’s hard to know and see what they’re doing away from here”. One coach cited the challenge of “what players do away from the club” when trying to develop physical qualities; “I don’t know what he’s doing regarding his diet. I get data back saying he’s training really hard, he’s doing everything, but he’s fat! That’s what I find hard; is what they do away from here”

Although practitioners are not able to control players actions when they are not within the club environment, one coach describes how they can educate players on correct techniques and protocols: 

“We can’t control everything, what the players do away from the club. But what we can do is look after this environment and make it as good as we can to encourage, guide and mould a champion at home as well as at training”

Another coach states how time and control can negatively influence players potential. The coach further questions whether they as coaches do enough to allow players achieve their full ability:

“I’ll give you a scenario, there’s four players here who are not reaching their potential because they’re physically not fit enough. So, they’ve gone through periodisation from a training perspective to try and improve them from a fitness point of view, but their body breaks down the body can’t cope with it. They have back problems, and shin splints. That’s my biggest challenge, so I’m on the case of our sport science team to try and improve the fitness of our players and bless them they’re doing the best, but these young players bodies can’t cope. Massive challenge that, and I find it very frustrating; seeing players leave the academy at the end of three years having not fulfilled their potential as their lifestyle and physical characteristics let them down. Did we do enough as coaches to help them reach their potential?”

A further sub-theme of external challenges that emerged from both coach and S&C coach interviews was speed. Speed was identified as the hardest physical quality for academy rugby league players to develop according to practitioners (coach; 67%, S&C coach 78%). One coach cited the influence of “genetics” on speed performance and the difficulty of significantly improving such qualities; “Speed—without doubt. Someone’s either quick or they’re not, you’re not going to turn them from a tortoise into a rabbit”. The notion of “it’s what you’ve got, really” and “it’s natural” were cited numerously, with one coach stating; “Speed due to genetics. I’m not sure how much you can change it; I don’t think you can make slow kids fast. I think you can change it slightly with technique and power but not hugely”

The sub-theme of time is linked to some of the challenges S&C coaches face when developing speed qualities. The lack of training time and “adequate rest” given to exclusively focus on speed development was cited as one of the external challenges faced by 60% of S&C coaches. 

“Speed and sprint performance is the most difficult to develop. Because, rest is required really, and coaches understanding of this. There is too much focus on mass development, and players are pre-fatigued (going) into sessions, (as well as) inappropriate recovery behaviours”

The concept of time was highlighted again by another S&C coach:

“Speed—the time it takes to do it properly. You don’t have adequate time in-season. You can’t really work on it in-season, it’s hard. It’s frowned upon by coaches, because they don’t understand the time or rest needed. I mean if you want to do… Say if you want your backs to do like 50–60 m so they get to top end speed, then you’ll need it (rest)… well, a large amount of it anyway. Technically, they say 1 min (rest) for every 10 m”

Coaches and S&C coaches views on the hardest physical quality to develop were similar, however, the reasons behind the difficulty was based on different reasons. It’s clear that coaches felt speed is wholly determined by genetic factors that are out of their control, while S&C coaches believe they are not afforded adequate time to develop speed qualities. In summary, it is evident that developing academy rugby league players is a multifaceted process with a plethora of factors associated with fitness testing and monitoring and developing physical qualities.

The current findings display the complex, multifaceted processes that are involved in developing, monitoring and evaluating physical qualities of academy rugby league players. Numerous factors present challenges for both coaches and S&C coaches regarding player’s physical development. The results suggest that practitioners working in academy rugby league would profit from the making of best practice guidelines.

### 2.10. Applied Implications

Our findings present novel insights from applied rugby league practitioners regarding their use, analysis and perceptions of fitness testing, alongside the development (and associated challenges) of physical qualities. Though the results are multifaceted, there are a number of suggestions that may be beneficial in supporting the development of academy rugby league players. Moreover, the themes could enhance the practices of coaches, S&C coaches, sport scientists and parents of players. The theme of “its important, but it’s not everything” would suggest rugby league practitioners to continue implementing fitness testing to objectively measure performance, but to also keep in mind the various factors associated with fitness data. The coach and S&C coach should have an understanding of each other’s perceptions of fitness testing, physical qualities and challenges, and from that, should work in unison to help counter the difference of opinions. For example, speed was identified as the hardest physical quality to develop, in future, coaches and S&C coaches could discuss this and related factors to ensure adequate training time is allocated to develop the quality. The head coach or manager could define a “performance model” and share with the S&C coach to achieve positive integration [62,63]. A novel, interactive method of visualising and monitoring fitness test results is warranted to ensure coaches and S&C coaches are presented with quick, easy to use but meaningful information regarding their players physical fitness. Moreover, fitness data should be compared to national norms to facilitate player development and provide an increased motivation stimulus for players. Conducting fitness testing at various timepoints is advised in order to simplify decision making, inform training programmes, and encourage motivation to successfully improve academy rugby league players. With regard external challenges, and especially parents, increased work and effort could be made to improve their understanding of long-term athletic development. Furthermore, a clear communication dialogue between coach and parent could enhance player progression.

### 2.11. Limitations and Future Research Directions

Whilst this study is the first to examine practitioners’ perspectives on the use, analysis and perceptions of fitness testing it is not without its limitations. It is imperative to acknowledge that the view of all participants involved were based on their own beliefs, ideals and experiences within a rugby league academy in England. Moreover, all participants were males which may have influenced perceptions. Therefore, our findings may not replicate the views of all rugby league practitioners. As a result, the transferability of the findings to player development in other sports and countries is unknown. Consequently, practitioners should bear this in mind when applying findings in other settings. It must also be acknowledged that rugby league performance is not specifically focussed solely on physical qualities and fitness testing. A plethora of various factors influence academy rugby league player’s development and performance [64]. Concentrating on practitioners’ perceptions of fitness testing only represents a small, but important, element of a multidimensional process to enhance practitioners’ practice. A study investigating talent identification and physical development in other sports could create best practice guidelines for coaches and S&C coaches on how to adequately support rugby league players from a physical perspective.

## 3. Conclusions

Overall, rugby league practitioners believe that the use of fitness testing and monitoring of physical qualities is important in rugby league, and there are numerous positive implications of its presence. Fitness data can provide useful context regarding player’s physical development, current physical state, positively impact the training environment, while informing training and assisting in decision making processes. However, there appears to be differences in coach and S&C coaches’ perceptions and use of fitness data, which emphasises the difficulties surrounding academy rugby league player’s development and the complexity of working in a multidisciplinary team. Indeed, the unpredictability and individual nature of both fitness testing and rugby performance were highlighted by practitioners and must be considered. Further, rugby league practitioners should acknowledge their colleagues’ perceptions with the overall aim of optimal player development. Given the demanding nature of professional rugby league, academy practitioners must continue to enhance player development strategies and should bear these complex issues in mind when making decisions based on fitness data amongst a range of other multidimensional factors.

## Figures and Tables

**Figure 1 sports-08-00130-f001:**
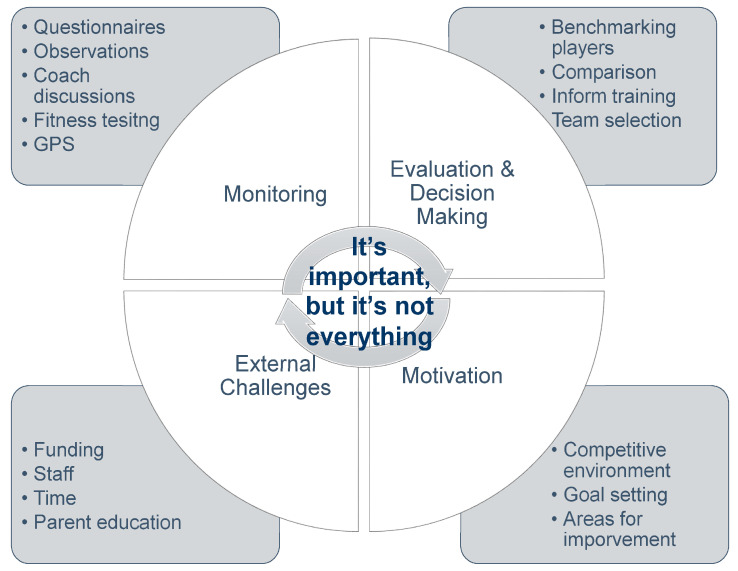
Thematic map of practitioner’s perceptions.

**Table 1 sports-08-00130-t001:** Practitioner response to questions regarding fitness data analysis, facilities and staff.

Question	Coach % (No.)	S&C Coach % (No.)
How do you analyse fitness data? *	By the S&C 71% (10)	Compare to previous data 80% (8)
	Don’t know 36% (5)	Statistical analysis 50% (5)
	Statistical analysis 21% (3)	
	Compare to previous data 14% (2)	
Do you have the necessary facilities/equipment to properly conduct fitness testing?	Yes 64% (9)	Yes 50% (5)
	Fair 29% (4)	Fair 30% (3)
	No 7% (1)	No 20% (2)

Do you have the adequate number of staff required?	Yes 71% (10)	No 60% (6)
	No 29% (4)	Yes 40% (4)

*** Some practitioners provided multiple answers.

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
