# Peer review of "“It’s Important, but It’s Not Everything”: Practitioners’ Use, Analysis and Perceptions of Fitness Testing in Academy Rugby League"

_sports, 2020, doi:10.3390/sports8090130_

Round 1

Reviewer 1 Report

Thank you very much for your interesting work and thank you very much to the journal for the opportunity to read it. I will now give my opinion on the paper with the aim that it can be improved.

The introduction talks in general terms about the importance of physical condition and gradually focuses on rugby.

It would be necessary for them to analyse more strictly the variables that are subsequently reflected in the semi-structured interviews and analysed in the results.
Likewise, the analysis of the state of the art of studies on the subject should be improved. They should make a clear statement of the most current studies on the subject by going through the studies in chronological order or by geographical areas.

With regard to the section on semi-structured interviews, the first paragraph could be part of the procedure.
It is very important to introduce in this section, the procedure for validating the content of the semi-structured interviews.

The results should be further synthesized and kept to those that respond directly to the objectives of the research and can be compared with those of other authors. All other results should be discarded.

As for the conclusions, they do not clearly respond to the objectives. They speak of perceptions and only in the first sentence does it seem to conclude about opinions. They should make this clearer.

Reviewer 2 Report

It was pleasure to read this paper. Research topic is really related with very important issue of the sport.
Although the study has much potential, there are also several limitations that need to be addressed before the manuscript could be recommended for publication.
The abstract and introductory part in general is well written, but I have a methodological question - is it possible to compare perceptions between coach and S&C coach when is choosing the methodology of qualitative research?
The Methods part needs corrections. First of all, please state the philosophical position of the research and explain the ontological and epistemological assumptions and how they guided the research. Second, this part lacks a methodological rigor section.
Also, in my opinion, the quality of the article would be improved by separate parts of the findings and discussion.
The study concludes that „... there appears to be some differences in coach and S&C coaches’ perceptions and use of fitness data, which emphasises the difficulties surrounding academy rugby league player’s development and the complexity of working in a multidisciplinary team „ (line 588-590). A debatable or qualitative study is capable of determining this.
Particular attention should be paid to references - sources should be provided as required.

Round 2

Reviewer 1 Report

Thank you for yours changes

Author Response

Thank you again for your comments and review. 

Reviewer 2 Report

Dear authors, 

Thank you for outlining the changes you have made in the manuscript. The manuscript has improved from the previous version.

However, part of the references still require adjustments.

Author Response

Thank you again for your comment. 

The reference style has been updated and is now the MDPI style as recommended by the journal. The authors would like to thank you for highlighting this issue.